# Analysis of Thermal Insulation Thickness for a Container House in the Yanqing Zone of the Beijing 2022 Olympic and Paralympic Winter Games

**DOI:** 10.3390/ijerph192416417

**Published:** 2022-12-07

**Authors:** Yurou Tong, Hui Yang, Li Bao, Baoxia Guo, Yanzhuo Shi, Congcong Wang

**Affiliations:** 1School of Environment and Energy Engineering, Beijing University of Civil Engineering and Architecture, Beijing 100044, China; 2Beijing Municipality Key Laboratory of Heating, Gas Supply, Ventilating, and Air Conditioning Engineering, Beijing University of Civil Engineering and Architecture, Beijing 100044, China; 3School of Humanities, Beijing University of Civil Engineering and Architecture, Beijing 100044, China

**Keywords:** container house, insulation materials, energy efficiency, carbon emission, economic analysis

## Abstract

A large number of temporary housings (THs) were used in the Yanqing zone of the Beijing 2022 Olympic and Paralympic Winter Games. Taking a kind of container house (CH) used in Yanqing zone as a model, the objective of this paper is to analyze the effect of insulation thickness on heating energy consumption and corresponding carbon emission. The effect of service life of THs on economic thickness was also discussed. The simulation model was developed using EnergyPlus and the heating energy consumption with different insulation materials was simulated based on the meteorological parameters of the top of Xiaohaituo Mountain (2177.5 m) and the Olympic/Paralympic Village (950 m) in Yanqing zone. In the simulation process, the thermal insulation performance of the CH was enhanced with reference to the requirements of GB/T 51350-2019 Technical Standard for Nearly Zero Energy Buildings (NZEB) on one hand. Additionally, the insulation performance was evaluated in terms of payback period and carbon emission. On the other hand, the economic thickness of different insulation materials (rock wool (RW), extruded polystyrene (XPS), polyurethane (PU)) and the high performance vacuum insulation panel (HVIP)) for different service lives of CH was studied. Results show that the *U*-values of the envelope meeting the NZEB standard can decrease approximately 21.4–32.8% of the heating energy consumption, compared with the original envelope. When the service life of CH is extended to 20 years, the carbon emission is reduced by 18.5% and 29.5%. The payback period of HVIP is longest, up to 31.4 a, and the results of economic thickness show that when the service life of the CH ranges from 1 year to 20 years, the economic thickness range of RW is 47–235 mm, XPS is 41–197 mm, PU is 33–149 mm and HVIP is 4–18 mm at the altitude of 2177.5 m. At the altitude of 950 m, the economic thickness range of RW is 28–158 mm, XPS is 26–131 mm, PU is 25–118 mm, and HVIP is 2–12 mm. From an economic point of view, the service life of a CH has a significant impact on the choice of insulation thickness.

## 1. Introduction

Temporary housings (THs), as a special type of building, have been widely used for refuging survivors of natural disasters [1], for sheltering patients during widespread disease, for residency of constructors in large engineering projects or for accommodating people in great public activities. The lifetime of TH is usually from 6 months to 5 years [2], or even longer depending on its function and quality. THs are generally prefabricated with lightweight structures and easy to be built, dismantled and stored for future reuse [3,4]. Compared to constructing permanent buildings, the utilization of THs can not only progress the construction but also improve the energy efficiency and decrease carbon dioxide emission on these special occasions. Container houses (CHs), as a kind of THs, are increasingly and widely used, which originated from the reuse of shipping containers in architecture [5]. They are manufactured in standard dimensions, which makes them an excellent modular building [6]. However, some problems with the indoor environment are found for THs including for CHs. Air quality is recognized as an important factor having a major effect on human health and has been studied by many scholars [7,8,9]. For temporary shelters, there are survey results showing that the total concentration of Volatile Organic Compounds (VOCs) and Particulate Matter (PM) in the room is very high [10]. While more problems are focused on the indoor thermal environment. Chen et al. [11] found that poor sound and heat insulation are common problems for THs. Some scholars studied the indoor thermal environment of THs in cold regions and hot summer and cold winter zones in China, finding that the natural indoor temperature sometimes was not within the comfort range of the human body [12,13,14]. The occupants’ satisfaction with the thermal insulation performance of CHs is predominantly low [15]. Thapa et al. [16,17] conducted a series of surveys on the indoor thermal environment and thermal comfort in four main earthquake affected districts in Nepal and analyzed the total heat loss coefficient per floor area of five shelters in Lalitpur during winter, which ranged from 11.3 to 15.2 W/(m^2^·K), suggesting that thermal insulation was very low.

While the thermal performance of temporary buildings is receiving a lot of attention, the thermal technology is also improving. Honma [18] introduced that according to the special specification of the Hokkaido government, glass wool insulation of 100 mm should be applied in the wall and ceiling of emergency temporary housing. Soga et al. [19] mentioned that different solutions on thermal technology of THs had be taken in different regions of Japan, such as increasing the thickness of insulation in Hokkaido. In the Mediterranean area, Cross-Laminated Timber (CLT) technology was applied to THs [20,21], which provides high insulation and good air tightness. It enables building structures to avoid thermal bridging and allows achieving remarkable energy savings in the heating season [22]. For CHs, Grębowski et al. [23] proposed that the envelope can be internally insulated, using glass wool or insulated with sprayed polyurethane foams. Additionally, Berbesz et al. [24] also pointed out that proper insulation is an important aspect for container house design.

In cold and severe cold regions, it is important to improve the insulation performance of the envelope of THs. Mario et al. [25] proposed that increasing the thickness of the insulation was required to achieve the adequate thermal transmittance coefficient that corresponds to the location of the deployment of the troops by comparing the energy efficiency requirements of the tents and containers that were used in military camps and the energy-efficient design requirements that were demanded by the energy efficiency standards for buildings in the civil sector. A simulation study by Liu et al. [26] on increasing the different thickness of rock wool (RW) to the walls of a TH at a construction site in Tianjin showed that the average indoor temperature increased with increasing insulation thickness. Yang et al. [27] analyzed the energy performance of portable houses composed of loess and porous materials in different climate zones and showed that the benefits of increased insulation were higher in areas where heating energy was dominant, and an insulation thickness of at least 100 mm is recommended. Furthermore, in Cornaro et al.’s study [28], the average heating power demand decreased by 59.4% when they incorporated aerogel into the walls and roof as well as used the modular floor consisting of polyethylene grids to enhance the insulation performance of an emergency shelter tent.

In order to reduce energy consumption in buildings, many countries have introduced measures to improve their building energy-efficiency standards and increasingly deployed the zero/low energy buildings. The zero energy container units were investigated based on a variety of energy efficiency technologies, where insulation is always a crucial consideration [29,30,31]. Meanwhile, the heat transfer coefficient (*U*-values) of the envelope in these studies is designed to be very low (0.1 or 0.26 W/(m^2^·K)), and Vacuum Insulation Panels (VIPs) are mostly used, while considering the interior space of the CH. Yet, both the improvement of energy-efficiency standards and the deployment of low-energy buildings typically focus on reducing the energy use in building operation [32]. It is also important to consider the contribution of insulation materials to the life cycle environmental impact. Pargana et al. [33] evaluated the environmental impacts on the production of conventional thermal insulation materials (extruded and expanded polystyrene (XPS and EPS), polyurethane, expanded cork agglomerate and expanded clay lightweight aggregates (LWA)) by means of a cradle to gate LCA methodological approach, and concluded that different materials contribute differently to the environmental impact, with a low contribution in EPS to all impact categories and the largest impact of LWA on global warming potential. Llantoy et al. [34] developed a comparative life cycle assessment (LCA) of different insulation materials (polyurethane, XPS, and mineral wool) in the Mediterranean continental climate, whose results showed that XPS presented the worst environmental performance, and the best environmental performance was for the mineral wool and the cubicles with three insulation materials achieving an energy saving of 27%, 25%, 23%, respectively, in comparison to the non-insulated cubicle. Tettey et al. [32] improved the original design of a residential building to achieve reference buildings to different energy-efficiency levels and at the same time varied the insulation materials in different parts of the reference buildings to achieve hypothetical options with similar energy-efficiency levels. Furthermore, they compared the production primary energy and CO_2_ emission of the reference and optimum designs of the buildings under the different energy-efficiency standards. The results showed a reduction of approximately 6–7% in primary energy use and 6–8% in CO_2_ emission when the insulation material in the reference buildings is changed from RW to cellulose fiber in the optimum versions. On the other hand, economic rationality is also one of the factors that must be considered in the design of energy-saving buildings. With these consideration, both environmental sustainability and cost should be highly concerned in selecting the insulation material of a building. Annibaldi et al. [35] considered environmental and economic factors to develop a framework to identify the optimal material to be used to achieve the highest level of energy efficiency in building retrofits, and the research was applied to an industrial factory in Italy. The optimal thickness of ten materials with different origin were obtained and their results showed that all materials analyzed displayed economical savings, and three of them showed reductions in the production of emissions from an environmental point of view. Considering that the increase in insulation thickness makes the cost of insulation higher, some studies used life cycle cost (LCC) to evaluate the cost and benefit of the envelope and to find the optimal thickness of insulation material [36,37,38,39]. However, most researches of the assessment of envelope insulation at present were aimed at permanent buildings, rather than THs.

In this paper, the geometric model was created based on a CH that was used in the Yanqing zone of the Beijing 2022 Olympic and Paralympic Winter Games. We resorted to EnergyPlus to simulate the indoor air temperature and validated the simulation results with the measured values tested in the Yanqing zone. Then, the heating energy consumption of CH with different insulation materials was simulated under the meteorological parameters of the Yanqing zone. In the simulation process, we enhanced the thermal insulation performance of the CH with reference to the limit values of the *U*-values of the envelope according to energy-efficiency standards of permanent buildings. Environmental and economic benefits were analyzed for the production stage of insulation materials and the operation stage of the building. Furthermore, considering that the service life of the CH variables in the application situation, the economic thickness of the insulation materials with a service life of 1 to 20 years was analyzed using the life cycle cost method.

## 2. Methodology

### 2.1. Environmental Conditions

Yanqing National Alpine Skiing Centre is located in the south of Xiaohaituo Mountain, which has a wide range of altitude variations and a distinct vertical distribution of climate. The altitude of the Yanqing Olympic/Paralympic Village is approximately 950 m, while the altitude of the Top Starting Area is approximately 2177.5 m. Figure 1 shows the monthly mean temperature of the two altitudes and of the typical year in Beijing. Compared to the typical year in Beijing, the monthly mean temperature ranged from 7.3 to 17.1 °C lower at the altitude of 2177.5 m and from 0.9 to 6.6 °C lower at the altitude of 950 m. It can be seen that the hottest monthly average temperature at both sites did not exceed 21 °C, and the lowest average daily temperature was −36.2 °C at the altitude of 2177.5 m and −24.2 °C at the altitude of 950 m. Heating degree days (HDD18) represents the sum of the daily difference between the conventional heating temperature of 18 °C and the average monthly outdoor air temperatures according to *GB 50176-2016 Code for Thermal Design of Civil Building* [40]. In addition, the HDD18 were calculated to be 6520 °C·d and 3748 °C·d for the two altitudes, indicating a high heating demand. Since the daily mean temperature at both sites did not exceed 26 °C, summer cooling was not considered. According to different heating degree days, the high altitude location (2177.5 m) belongs to severe cold A regions and the low altitude location (950 m) belongs to cold regions [40], while the two locations belong to very cold and cool regions, respectively, according to Table Annex1-4 of the ASHRAE Standard 90.1-2019 [41]. With reference to the regulations for the heating season in the severe cold region of Harbin and the cold region of Beijing, the heating energy consumption calculation periods for the two regions are determined in this paper: 20/10-20/4 (2177.5 m) and 15/11-15/3 (950 m), respectively.

### 2.2. Experimental CH and Model Validation

A large number of THs (tents, cabins and CHs) were used in the Yanqing zone of the Beijing 2022 Olympic and Paralympic Winter Games. Figure 2 shows the grandstand and CHs whose functions are Commentary Position, Broadcast Information office and so on of the Technical Finish Area. In this research, a CH with dimensions of 6.00 m (L) × 3.00 m (W) × 2.9 m (H) was studied, which was the Traffic service Staff Lounge at Road Two, as shown in Figure 3a. A window (2.67 m × 0.9 m) and a glazed door (1.47 m × 2.57 m) on a 3 m × 2.9 m wall, respectively (Figure 3b). The thermophysical parameters of envelope enclosure is demonstrated in Table 1 and Table 2.

A simulation model for the CH was established using EnergyPlus, as shown in Figure 3c. For the purpose of model validation, a field research of the CH was carried out. The indoor air temperature and internal surface temperature of glass door were measured. Figure 4 shows the position of temperature measuring points. Indoor air temperature was measured in the corner of the CH at a height of 0.45 m from the floor to avoid the effects of solar radiation and the measuring point of surface temperature was 1.3 m from the floor. The USB temperature and humidity recorder with an accuracy of ±0.1 °C and SMD temperature recorder with an accuracy of ±0.2 °C were used to measure air temperature and surface temperature, respectively, as shown in Figure 5a,b. In addition, weather parameters including outdoor air temperature, relative humidity, horizontal global solar radiation, horizontal diffuse radiation, atmospheric pressure, wind direction and wind speed were recorded by a meteorological station nearby, as shown in Figure 5c.

All the measuring instruments were calibrated before use. The measurement time interval for all parameters was 5 min. The measurement started on 1 October 2021 and ended on 10 October 2021 to collect experimental data for model validation. During the measurements, the indoor temperature was affected on 6 October by the entry of staff into the room, so the data from the 6th was excluded when comparing the actual measurement with the simulated data later. The door and the windows were closed at all other times of the measurements, and there were no occupants and heat-generation equipment inside the experimental CH.

The comparisons between the simulated and measured temperature are as shown in Figure 6. As seen, a good agreement between the simulated and measured indoor air temperatures was achieved. To further ascertain the validation accuracy, two basic statistical parameters were calculated. One was the Mean Absolute Error (MAE) and the other coefficient of Root Mean Square Error (RMSE), using Equations (1) and (2), as follows:
(1)$$MAE=\frac{1}{N}\sum_{i=1}^{N}|m_{i}-s_{i}|$$
(2)$$RMSE=\sqrt{\sum_{i=1}^{N}(m_{i}-s_{i}{)}^{2}/N}$$
where *m_i_* and *s_i_* are the respective measured temperature and simulated temperature for each time interval ‘*i*’, and *N* is the total number of hours used for validation.

The MAE and RMSE for air temperature are 0.98 °C and 1.22 °C, respectively, and are 1.23 °C and 1.69 °C for the surface temperature of the glass door, which are within the permissible error range [42]. Therefore, the accuracy of the simulation was verified with experiments. Then, the simulation method was utilized to calculate the heating energy consumption of the CH.

### 2.3. Numerical Simulation

#### 2.3.1. Simulation Parameters Setting

EnergyPlus was used to simulate the heating energy consumption of the CH, and the parameters were set according to the actual conditions of the Olympic and Paralympic Winter Games. The occupancy schedule is 07:00–17:00. The density of lighting is 4 W/m^2^, and the number of people in the room is 1. Given the function of the room, no consideration was given to electrical equipment, such as printers, computers, etc. No mechanical ventilation was set up, the airtightness of the doors and windows was class V [43], and the infiltration air volume was 0.01 m^3^/s, meeting the minimum fresh air volume of 30 m^3^/s required by personnel [44]. Electric heaters were used for heating, with an electric heat conversion efficiency of 90%, and the design heating temperature is 18 °C from 07:00 to 17:00, and 10 °C for the rest of the time.

In this research, we conducted the comparative simulations between the conventional insulation materials (rock wool (RW), extruded polystyrene (XPS) and polyurethane (PU)) and the high performance vacuum insulation panel (HVIP) of which core material is fumed silica. Related studies have proposed a linear temperature-dependent law that displays a decreased thermal conductivity at low temperatures [45,46,47]; the thermal conductivity of the HVIP was tested at different temperatures and the results showed 0.0039 W/m·K at −20 °C and 0.0048 W/m·K at 25 °C. Thus, in order to unify the temperature conditions, the thermal conductivity of the material at room temperature is used in this paper. Despite that, the results calculated at room temperature conditions will be larger than actual results since the actual environment has a low temperature, this is for the sake of safer consideration. The performance parameters of materials are shown in Table 3, where the parameters of HVIP are provided by qualified third-party testing institution and other materials are referred to [40,48,49,50].

#### 2.3.2. Insulation Thickness under the NZEB Standard

The original envelope of CH was improved with reference to the limits on the *U*-values of the envelope by permanent building standard *GB/T 51350-2019 Technical Standard for Nearly Zero Energy Buildings* (NZEB) [51]. Table 4 shows the *U*-values for severe cold regions and cold regions. It is noted that the *U*-values are within the range required by the standard, rather than the lower limit value in order to ensure that the thicknesses of the materials are all integers. Table 5 is insulation materials with required thicknesses in the different building parts under the NZEB standard. The walls, roof and floor of the original envelope are insulated with 75 mm RW.

### 2.4. Indicators of Economic and Environmental Analysis

Economic rationality is one of the factors that must be considered in the design of energy-saving buildings and is equally important for THs. Using original envelope as the baseline scheme, the financial payback effect of the energy efficiency schemes under the NZEB standard is measured by calculating the payback period. There are two methods to calculate the payback period: the simple payback and discount payback. However, simple payback period is usually shorter than the latter in an inflationary economic context since it is calculated by omitting the time value of currency. So, the discount payback period that is calculated by taking into account the time value of currency and converting the future energy savings into the present value is more likely to be in line with the economic reality. Hence, this paper calculated the discount payback period of each insulation material under NZEB standard by Equations (3) and (4) [52].(3)$$P_{D}=T^{\prime }-1+\frac{|\sum_{t=0}^{T^{\prime }-1}{\left(CI-CO\right)}_{t}(P/F,r,t)|}{{\left(CI-CO\right)}_{T^{\prime }}(P/F,r,T^{\prime })}$$
(4)$$(P/F,r,t)=\frac{1}{{\left(1+r\right)}^{t}}$$where *P_D_* is the discount payback period, *T*′ is the number of years with positive cumulative discounted value, *CI* represents cash inflow and its value equals to annual saving electric charge, *CO* represents cash outflow and its value equals to the increased cost of insulation, *P*/*F* is present-value compound interest factor and *r* is discount rate.

As the service life of a container house varies with application situation, it is necessary to determine the thickness of the insulation according to its service life. Life Cycle Cost (LCC) is one of the popular financial analysis methods, which has been used to evaluate the cost and benefit of the building envelope [53]. This analysis method considers the initial cost of the insulation materials plus the ongoing energy cost over the expected lifetime of the insulation [37]. Through the use of the LCC, the optimal isolation level may be defined keeping costs to a minimum. Equations (5) and (6) are the proposed LCC calculation expressions.(5)$$LCC=E_{H}\times E_{T}\times PVF\times (C_{P}+C_{I})\times c$$
(6)$$PVF=\frac{{\left(1+r\right)}^{N}-1}{r\times {\left(1+r\right)}^{N}}$$where *E_H_* is the heating energy consumption per unit area, which is simulated by EnergyPlus; *E_T_* is electricity tariff, which was based on the field investigation after the Paralympic Winter Games in Yanqing; *PVF* is the Present Value Function; *C_P_* is the price of thermal insulation; *C_I_* is the installation cost of the insulation material; *c* is the consumption of insulation materials; *N* is the service life for the CH in a place. The details of calculation parameters are shown in Table 6.

In this paper, the maximum service life of the CH is 20 years. For the long-term application of insulation materials, there are aging phenomena that exist. Relevant research related to the variation of thermal conductivity of insulation materials influenced by aging showed that the thermal conductivity of different insulation materials affected by multiple factors, such as material properties, production process level, and application conditions [60,61,62,63,64], while the variation of thermal conductivity of four insulation materials with the practical climate conditions has not been conducted in this paper. So, it was assumed that the thermal conductivity of the four materials is fixed in this paper.

Accurate accounting of building carbon emissions can promote the healthy development of green buildings and can enable them to truly achieve their original goal of energy saving and emission reduction. Scholars have developed process-based method, input–output method, and hybrid method for the emission assessment of buildings [65]. Process-based method combines activity data with relevant emission factors to assess the emissions and has been widely used in the analysis of individual buildings [66]. Therefore, the process-based method was chosen to calculate the carbon emissions. The total carbon emissions in this research include the production stage of insulation material and the operation stage of the CH, as shown in Equation (7):(7)$$C_{T}=M\times EF_{PRO}+W\times EF_{OPE}$$where *M* and *EF_PRO_* are the consumption and emission factor for insulation materials, respectively; *W* and *EF_OPE_* are the electricity consumption and electricity carbon emission factors, respectively. Although renewable energy was used in the Beijing 2022 Olympic and Paralympic Winter Games, the carbon emission factor of electricity issued by Ministry of Ecology and Environment of the People’s Republic of China was used to make the study more universal. Carbon emission factors are presented in Table 6.

## 3. Results and Discussions

### 3.1. High Insulation Performance CH

#### 3.1.1. Usable Area and Volume of the Room

From Table 5, it can be seen that when the U-values are within the range required by NZEB standard in severe cold region, the required thickness of RW can be as high as 200 mm, which is far beyond 25 mm, the maximum thickness of HVIP. HVIP insulation consistently gives the lowest thickness required. For the internal insulation of container houses, the thickness of the insulation material will greatly affect the internal usable area and volume of the room. Table 7 shows the usable area (S) and the usable volume (V) of the room under different schemes. With the improvement of insulation performance, both S and V drop steeply when conventional insulation materials are used for insulation due to the dramatic increase in material thickness. However, if HVIP is used, the usable area of the room when reaching the NZEB standard is also larger than the original envelope.

#### 3.1.2. Heating Energy Consumption

Table 8 shows the heating energy consumption based on both of the two meteorological parameters. The heating energy consumption is 32.8% and 21.4% lower than the original envelope at the altitude of 2177.5 m and 950 m, respectively, when the *U*-values of CH meet the NZEB standard. It is important to improve the insulation performance of CHs, from the point of view of building energy efficiency. The annual heating energy consumption per floor area under the NZEB standard was 193.3 kW·h/(m^2^·a) and 113.92 kW·h/(m^2^·a) on average at the altitude of 2177.5 m and 950 m, respectively. One of the main reasons for the high energy consumption is that CH, as a single building, has a shape factor of 1.69 in this paper, while GB 55015-2021 *General Code for Energy Efficiency and Renewable Energy Application in Buildings* [67] specifies a shape factor of 0.5 or less for public buildings in severe cold and cold regions. An excessively large shape factor will lead to a great heat dissipation on the external surface and more energy consumption.

#### 3.1.3. Economic and Environmental Assessment

The payback period for schemes under the NZEB standard was calculated as shown in Table 9. HVIP has long payback periods at both the altitudes. On one hand, the upfront investment cost of HVIP is high and, on the other hand, the energy savings in the operational phase of the CH are not significant especially for the altitude of 950 m. As a result, the savings in the operational phase of the CH take a longer time (even more than 20 years) to cover the upfront capital investment. It is revealed that the payback period of conventional insulation materials lasts less than 10 years.

Figure 7 shows the total carbon emissions of the CH at the altitude of 2177.5 m and 950 m, in terms of different service lives. Carbon emissions are reduced as the service lives of CH increase and energy efficiency standards improve. At the altitude of 2177.5 m, the reduction in carbon emissions increased from 13.6% to 29.5% compared to original envelope when the service life of CH is extended from 3 years to 20 years, and at the altitude of 950 m, the carbon emissions of cases under the NZEB standard are reduced by 15.6% and 18.5% at 10 and 20 years, respectively, compared to original envelope. So, from an environmental point of view, energy saving becomes equally important when CHs are used in the long term. The difference in total carbon emissions when CH is insulated with conventional and high performance insulation materials, respectively, is not significant, with a maximum difference of 0.9 t when the service life of CH is 20 years under the same meteorological parameter. Although the carbon emission factor of conventional insulation materials is seven times smaller than that of HVIP, its high dosage puts it at a disadvantage. On the other hand, the difference in heating energy consumption between the two under the same standard is not large either.

### 3.2. Economic Thickness

Figure 8 shows the variation of cost with insulation thickness for a container house with a service life of 5 years. As expected, the cost of heating energy consumption decreases with the increase in insulation thickness while the initial cost increases. The total cost decreases and then increases as the thickness of the insulation increases.

The minimum total life cycle cost of the four insulation materials and the corresponding insulation thickness for each year from 1 to 20 years were calculated, and, the variation of economic thickness and total cost with service life are presented in Figure 9. It can be seen that the service life of CH has a significant effect on economic thickness and total life cycle cost whose values increase with the length of the service life, showing a trend of rapid increase and then slow growth, which is consistent with the results in [68,69].

Therefore, from an economic point of view, the thickness of insulation materials can be selected according to the service life (1–20 years) of the CH: at the altitude of 2177.5 m, the economic thickness range of RW is 47–235 mm, with a total cost of 1–4.97 × 10^4^ RMB; the economic thickness range of XPS is 41–197 mm, with a total cost of 0.92–4.66 × 10^4^ RMB; the economic thickness range of PU is 33–149 mm, with a total cost of 0.93–4.73 × 10^4^ RMB; the economic thickness range of HVIP is 4–18 mm, with a total cost of 1.28–6.41 × 10^4^ RMB. At the altitude of 950 m, the economic thickness range of RW is 28–158 mm, with a total cost of 0.65–3.26 × 10^4^ RMB; the economic thickness range of XPS is 26–131 mm, with a total cost of 0.59–3 × 10^4^ RMB; the economic thickness range of PU is 25–118 mm, with a total cost of 0.55–2.8 × 10^4^ RMB; the economic thickness range of HVIP is 2–12 mm, with a total cost of 0.76–3.89 × 10^4^ RMB. It can be seen that the economic thickness of conventional insulation materials spans a wide range, especially for RW.

Meanwhile, the carbon emissions corresponding to the economic thickness of the four insulation materials were calculated, as shown in Figure 10. Due to the high price of HVIP, its economic thickness is much smaller compared to conventional insulation materials in the same year, while the heating energy consumption required will be larger than conventional insulation materials, so its carbon emission is higher than conventional insulation materials.

## 4. Conclusions

In this study, the insulation of a CH was improved with reference to the requirements of NZEB standard for thermal performance of the envelope, the heating energy consumption of CH with different insulation materials was simulated and their economic and environmental benefits were analyzed. Meanwhile, the economic thickness of different insulation materials was calculated considering the service life of the CH.

Improving the insulation performance of CH has a great energy-saving effect. The annual heating energy consumption of CH is reduced by 32.8% at the altitude of 2177.5 m and 21.4% at the altitude of 950 m compared to the original envelope when the *U*-values of the envelope meet the NZEB standard. In addition, carbon emissions are reduced as the service lives of CH increase. After reducing the *U*-values of the envelope to meet NZEB standard, carbon emissions are 18.5% lower at the altitude of 2177.5 m and 29.5% lower at the altitude of 950 m, if the CH is used for 20 years. Therefore, it is evidently seen that HVIP has the longest payback period, up to 31.4 years at the altitude of 950 m, which is less economically beneficial when meeting the NZEB standard.

What is more, the service life of the CH has a great impact on the economic thickness. When the service life is from 1 to 20 years, the economic thickness range is 47–235 mm for RW, 41–197 mm for XPS, 33–149 mm for PU, and 4–18 mm for HVIP at the altitude of 2177.5 m. At the altitude of 950 m, the economic thickness range is 28–158 mm for RW, 26–131 mm for XPS, 25–118 mm for PU, and 2–12 mm for HVIP. The total cost and carbon emission of the economic thickness of HVIP is still higher than that of conventional insulation materials. If HVIP is widely used in the future, the price may be reduced, which is likely to result in the reduction of the total cost and carbon emission reduced.

CHs are now a commonly used prefabricated element in the building industry. The CHs achieved a variety of functional applications during the Beijing 2022 Olympic Winter Games. After the game, as sustainable legacies, the CHs have been recycled and transformed as convenience stores, reading bars, hotel rooms and so on to play an important role in urban comprehensive development and urban renewal. Although the research of this paper is based on the meteorological parameters of Xiaohaituo Mountain, the results are also appropriate for similar climate conditions.

## Figures and Tables

**Figure 1 ijerph-19-16417-f001:**
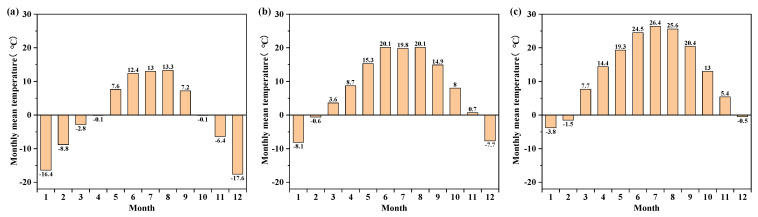
Monthly mean temperature (**a**) 2177.5 m; (**b**) 950 m; (**c**) typical year.

**Figure 2 ijerph-19-16417-f002:**
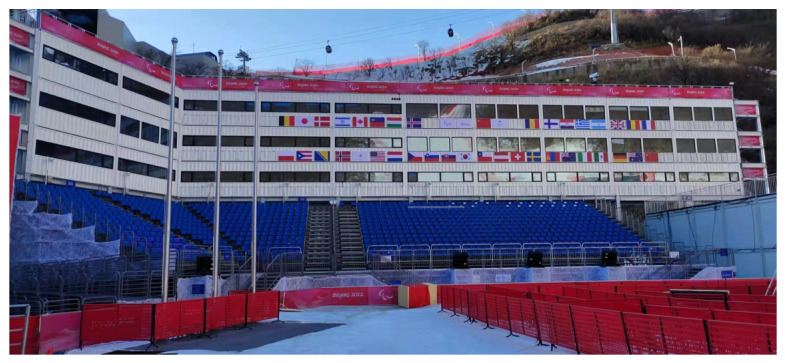
The grandstand and CHs of the Technical Finish Area.

**Figure 3 ijerph-19-16417-f003:**
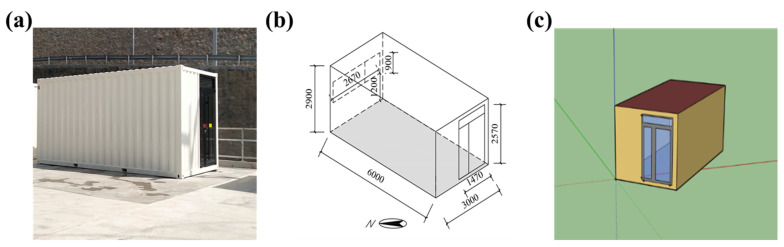
(**a**) The photo of the experimental CH; (**b**) The dimensions of the CH; (**c**) Simulation model.

**Figure 4 ijerph-19-16417-f004:**
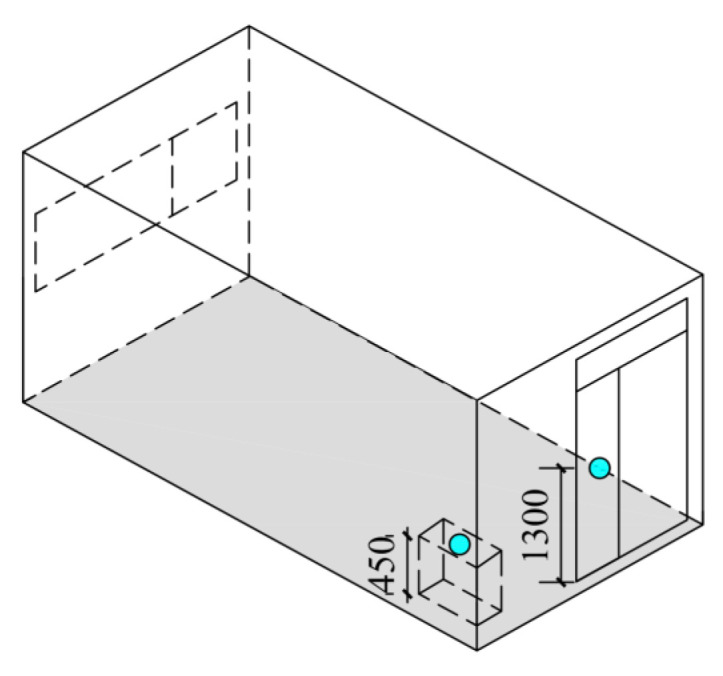
The location of the temperature measuring point in the experimental CH.

**Figure 5 ijerph-19-16417-f005:**
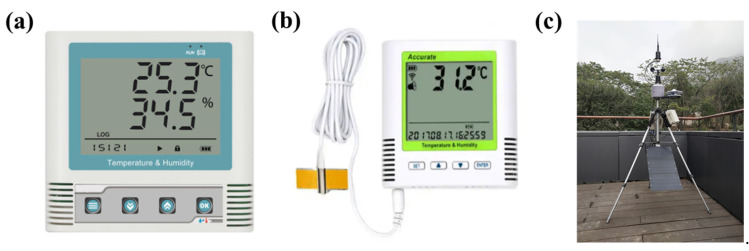
(**a**) USB temperature and humidity recorder; (**b**) SMD temperature recorder; (**c**) meteorological station.

**Figure 6 ijerph-19-16417-f006:**
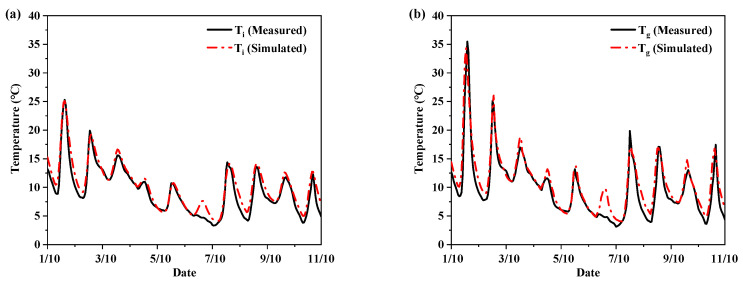
Comparisons between the measured and simulated (**a**) indoor air temperatures; (**b**) internal surface temperatures of glass door.

**Figure 7 ijerph-19-16417-f007:**
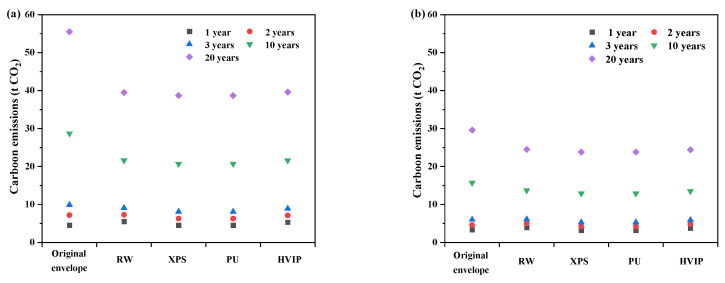
Carbon emissions of the original envelope and the optimized envelope, in terms of different service lives (**a**) 2177.5 m; (**b**) 950 m. Note: The *U*-values of the optimized envelope meet the NZEB standard.

**Figure 8 ijerph-19-16417-f008:**
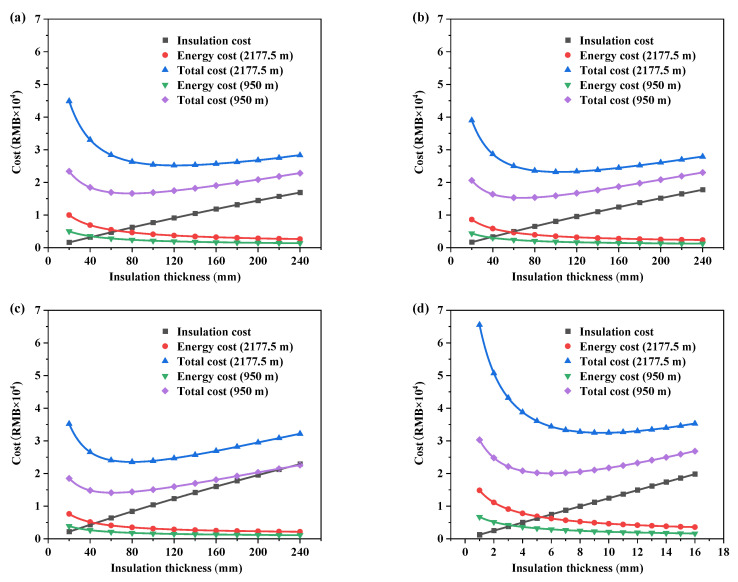
Variation of cost with insulation thickness (**a**) RW; (**b**) XPS; (**c**) PU; (**d**) HVIP.

**Figure 9 ijerph-19-16417-f009:**
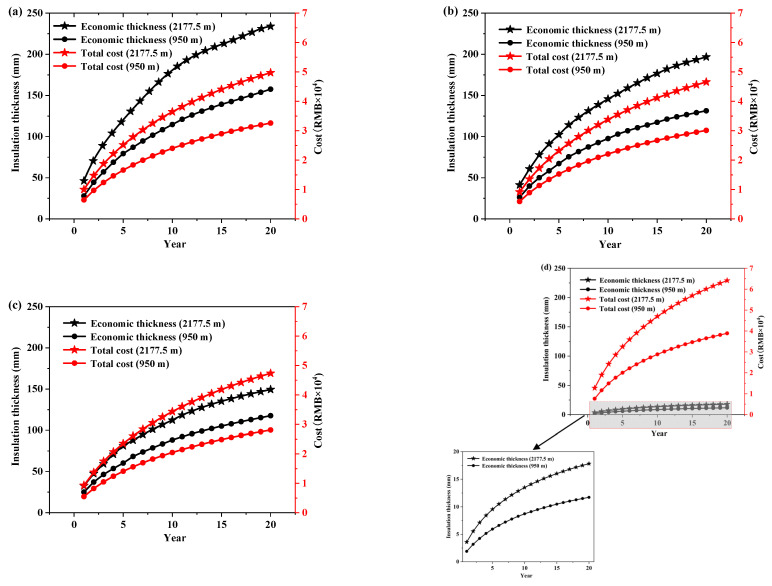
Variation of economic thickness and total cost with service life (**a**) RW; (**b**) XPS; (**c**) PU; (**d**) HVIP.

**Figure 10 ijerph-19-16417-f010:**
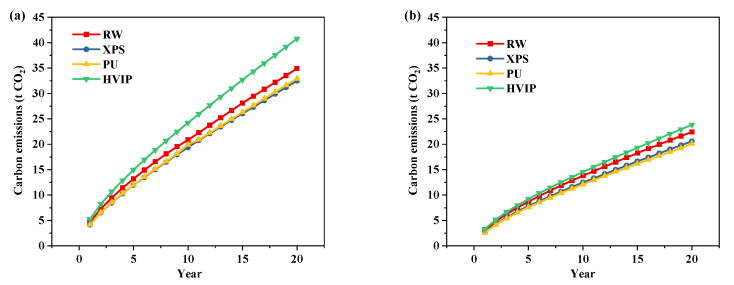
Carbon emissions corresponding to the economic thickness (**a**) 2177.5 m; (**b**) 950 m.

**Table 1 ijerph-19-16417-t001:** Thermophysical parameters of non-transparent envelope.

Envelope	Structures	Thickness mm	Thermal Conductivity W/(m·K)	Specific Heat J/(kg·K)	Density kg/m^3^
Wall	Steel panel	1.2	45	300	7800
	RW	75	0.04	1220	160
	Aluminum honeycomb panel	9	0.88	300	10
Floor	Steel panel	1.2	45	300	7800
	RW	75	0.04	1220	160
	Cement pressure plate	18	0.039	900	350
	reflective coating	1	0.039	837	2710
	Composite wood board	12	0.24	1500	800
Roof	Steel panel	1.2	45	300	7800
	RW	75	0.04	1220	160
	Aluminum honeycomb panel	5	0.88	300	10

**Table 2 ijerph-19-16417-t002:** Thermophysical parameters of transparent envelope.

Envelope	Structure	Heat Transfer Coefficient of Glass W/(m^2^·K)	Solar Heat Gain Coefficient (SHGC)
Window/Door	6 mm Low-E glass	1.8	0.43
	12 mm air layer		
	6 mm clear glass		

**Table 3 ijerph-19-16417-t003:** Characteristics of the selected insulation materials.

Insulation Material	Thermal Conductivity (W/m·K)	Specific Heat (J/kg·K)	Density (kg/m^3^)	Fire Rating
RW	0.04	1220	160	A
XPS	0.03	1380	35	B1
PU	0.024	1380	35	B1
HVIP	0.005	876	200	A

**Table 4 ijerph-19-16417-t004:** *U*-values of envelope under the NZEB standard.

Building Part	*U*-Value (W/m^2^·K)	
	Severe Cold Region	Cold Region
Wall	0.24	0.298
Roof	0.194	0.299
Floor	0.3	0.395

**Table 5 ijerph-19-16417-t005:** Insulation material thicknesses for different parts of envelopes required under the NZEB standard.

Building Part	2177.5 m				950 m			
	RW	XPS	PU	HVIP	RW	XPS	PU	HVIP
Wall	160	120	96	20	128	96	77	16
Roof	200	150	120	25	128	96	77	16
Floor	104	78	62.5	13	72	54	43	9

**Table 6 ijerph-19-16417-t006:** Calculation parameters.

Parameter	Value		Unit
*C_P_*	RW	750	RMB/m_3_
	XPS	800	
	PU	1100	
	HVIP	15,000	
*C_I_*	240		RMB/m_3_
*E_T_*	1.05		RMB/kWh
*N*	1~20 [30,54]		year(s)
*r*	5% [53,55]		-
*EF_OPE_*	0.5839 [56]		kgCO_2_/(kW·h)
*EF_PRO_*	RW	316.8 [57]	kgCO_2_/m_3_
	XPS	296.6 [58]	
	PU	363.7 [58]	
	HVIP	2220 [59]	

**Table 7 ijerph-19-16417-t007:** Usable area (S) and usable volume (V) under the NZEB standard.

S (m^2^)/V (m^3^)	Original Envelope	2177.5 m				950 m			
		RW	XPS	PU	HVIP	RW	XPS	PU	HVIP
S	16.7	15.2	15.9	16.3	17.6	15.8	16.3	16.6	17.7
V	45.8	39.5	42.5	44.3	50.5	42.6	44.8	46.3	50.9

The thickness corresponding to different insulation materials are shown in Table 5.

**Table 8 ijerph-19-16417-t008:** The heating energy consumption.

	2177.5 m					950 m				
	Original Envelope	RW	XPS	PU	HVIP	Original Envelope	RW	XPS	PU	HVIP
Energy consumption (kW·h)	4588.9	3065	3087.9	3088.1	3089.5	2372.7	1857.6	1866	1866.4	1867.7

The thickness corresponding to different insulation materials are shown in Table 5.

**Table 9 ijerph-19-16417-t009:** The payback periods.

	2177.5 m				950 m			
	RW	XPS	PU	HVIP	RW	XPS	PU	HVIP
Payback period (year)	3.96	2.5	2.8	13.6	6.4	2.9	3.5	31.4

The thickness corresponding to different insulation materials are shown in Table 5.

## Data Availability

All data generated or analyzed during this study are included in this published article.

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
