# Peer review of "Analysis of Thermal Insulation Thickness for a Container House in the Yanqing Zone of the Beijing 2022 Olympic and Paralympic Winter Games"

_ijerph, 2022, doi:10.3390/ijerph192416417_

Round 1

Reviewer 1 Report

In this paper, a simulation model of a container house (CH) was developed. By considering different insulation materials, the heating energy consumption of CH was simulated and the insulation performance was evaluated in terms of payback period and carbon emission. The economic thickness of different insulation materials for different service lives of CH was also compared and analyzed. Overall, the paper is well written. I just have a few comments to the authors:

1.      Figure 6, why there are obvious inconsistencies between the 6th and 7th day?

2.      What is the dimension of HVIP? Does it require any sealant among different pieces and which type of the sealant is? After applying the sealant, what is the overall thermal conductivity?

3.      The authors mentioned “Due to the lack of data on the predicted thermal performance of the HVIP in this paper and the fact that the thermal conductivity of the VIPs varies considerably in existing studies, which depends on the production process and the level of technology, it is assumed then that the thermal conductivity of the HVIP is still 0.005 W/m·K when the CH is applied over a long period of time in this paper and it is expected that the performance of VIP will not degrade significantly as the technology develops.” Then why not consider a correction factor during the following analysis and update the economic thickness range.

4.      How about the aging affect, or deteriorations of rockwool, XPS and PU?

Author Response

Detailed responses to reviewers’ comments of manuscript ijerph-2039407

Analysis of thermal insulation thickness for container house in Yanqing Zone of Beijing 2022 Olympic and Paralympic Winter Games

Dear Editor and reviewers,

Thank you for your thorough review and constructive comments on our manuscript! We seriously considered the comments and carefully revised the manuscript. In addition, we have made some modifications on English expression in this revision.

All the changes in the revision were marked up using the “Track Changes”. We believe most of the comments have been addressed in the revised manuscript. The point-by-point response to the reviewers’ comments are listed below. Hope this work can be accepted to publish in International Journal of Environmental Research and Public Health.

Reviewer 1

In this paper, a simulation model of a container house (CH) was developed. By considering different insulation materials, the heating energy consumption of CH was simulated and the insulation performance was evaluated in terms of payback period and carbon emission. The economic thickness of different insulation materials for different service lives of CH was also compared and analyzed. Overall, the paper is well written. I just have a few comments to the authors:

1. Figure 6, why there are obvious inconsistencies between the 6th and 7th day?

The reason for the inconsistency is that staff entered the container house on the 6th day, which affected the measured results of the indoor temperature and led to a significant difference between the measured and simulated values. And, this reason was explained in our manuscript, on page 6, from line 208 to line 211, as “During the measurements, the indoor temperature was affected on 6 October by the entry of staff into the room, so the data from the 6th was excluded when comparing the actual measurement with the simulated data later.”.

2. What is the dimension of HVIP? Does it require any sealant among different pieces and which type of the sealant is? After applying the sealant, what is the overall thermal conductivity?

The standard dimensions of HVIP are 600 mm (L) and 400 mm (W). Non-standard dimensions are prefabricated, i.e., calculated according to the actual conditions and customized by the manufacturer.

The gap between the different pieces is less than 2 mm. And for the installation of multi-layer, stagger the gap of the previous layer of HVIP to reduce the problem of thermal bridging caused by the gap. Therefore, no sealant was applied. 

3. The authors mentioned “Due to the lack of data on the predicted thermal performance of the HVIP in this paper and the fact that the thermal conductivity of the VIPs varies considerably in existing studies, which depends on the production process and the level of technology, it is assumed then that the thermal conductivity of the HVIP is still 0.005 W/m·K when the CH is applied over a long period of time in this paper and it is expected that the performance of VIP will not degrade significantly as the technology develops.” Then why not consider a correction factor during the following analysis and update the economic thickness range.

Thank you for the comment! Considering the correction factor is really a good idea. And, we consulted the relevant references related to assessing the performance of VIPs and studying the variation of thermal conductivity with service life for VIPs. Johansson et al. [1] conducted an experimental investigation of a building retrofitted with VIPs in Switzerland over a period of 5 years and the measurements of temperature and relative humidity in the wall showed no sign of deterioration of the VIP. In Canada, Molleti et al. [2] monitored the VIP composite in-situ for five years and found that the long-term performance of VIP composite performed within a 10% margin of its original value for consecutive years. Batard et al. [3] showed that VIPs with hydrophobic core materials have better long-term thermal performance than those with hydrophilic core materials. For example, it is predicted that thermal conductivity changed from approximately 0.00385 W/m·K to 0.00488 W/m·K for the former and from 0.0035 W/m·K to 0.00605 W/m·K for the latter over 20 years when installed on a flat roof of Nice. The study also showed that when VIPs are used for a longer period of time (50 years), their thermal conductivity depends more on the climate and even more on the core material. Not only that, the barrier envelope also affects the insulation performance of VIP. Mao et al. [4] predicted thermal conductivity of three VIPs with the different barriers and showed that the VIPs with the barrier MF1 increased from approximately 0.0026 to 0.0063 W/m·K and the VIPs with the barrier laminate AF varied from 0.0022 to 0.0033 W/m·K when the service life is 20 years. It can be seen that the variation of thermal conductivity of VIP is influenced by multiple factors such as material properties, production process level, and application conditions. At present, we have not conducted relevant work on the variation of thermal conductivity of HVIP with the service life, so it is difficult to determine the correction factor. But it is worth studying in the future.

[1] Johansson, P., Adl-Zarrabi, B., Sasic Kalagasidis, A., 2016. Evaluation of 5 years’ performance of VIPs in a retrofitted building façade. Energy and Buildings 130, 488–494.

https://doi.org/10.1016/j.enbuild.2016.08.073

[2] Molleti, S., Lefebvre, D., van Reenen, D., 2018. Long-term in-situ assessment of vacuum insulation panels for integration into roofing systems: Five years of field-performance. Energy and Buildings 168, 97–105. https://doi.org/10.1016/j.enbuild.2018.03.010

[3] Batard, A., Duforestel, T., Flandin, L., Yrieix, B., 2018. Prediction method of the long-term thermal performance of Vacuum Insulation Panels installed in building thermal insulation applications. Energy and Buildings 178, 1–10. https://doi.org/10.1016/j.enbuild.2018.08.006

[4] Mao, S., Kan, A., Zhu, W., Yuan, Y., 2020. The impact of vacuum degree and barrier envelope on thermal property and service life of vacuum insulation panels. Energy and Buildings 209, 109699. https://doi.org/10.1016/j.enbuild.2019.109699

4. How about the aging affect, or deteriorations of rock wool, XPS and PU?

Rock wool, XPS and PU can also suffer from ageing due to ambient temperature, humidity and other factors. Scholars have also carried out research in this field, mainly conducting experiments or simulations at different temperatures and humidity levels to investigate the changes in thermal conductivity [5-7]. However, at present, it is difficult to determine how the thermal conductivity of insulation materials will change due to the variation of indoor and outdoor environment of the container house in this paper. Moreover, this article focuses on taking the CH as a model to analyze the economic and environmental benefits of insulation materials and the economic thickness ranges. So, the variation of thermal conductivity considering the aging effect was not considered in this paper and it is worth conducting relevant work in the future.

Thank you for this question! We realized that the original manuscript was unclear about this section, so we have made some modifications, on page 10, from line 309 to line 316, as “In this paper, the maximum service life of the CH is 20 years. For the long-term application of insulation materials, there are aging phenomena that exist. Relevant research related to the variation of thermal conductivity of insulation materials influenced by the aging showed that the thermal conductivity of different insulation materials affected by multiple factors such as material properties, production process level, and application conditions [60-64]. While, the variation of thermal conductivity of four insulation materials with the practical climate conditions has not been conducted in this paper. So, it was assumed that the thermal conductivity of the four materials is fixed in this paper. ”.

[5] Wang, Y., Liu, K., Liu, Y., Wang, D., Liu, J., 2022. The impact of temperature and relative humidity dependent thermal conductivity of insulation materials on heat transfer through the building envelope. Journal of Building Engineering 46, 103700. https://doi.org/10.1016/j.jobe.2021.103700

[6] Zhang, H., Shang, C., Tang, G., 2022. Measurement and identification of temperature-dependent thermal conductivity for thermal insulation materials under large temperature difference. International Journal of Thermal Sciences 171, 107261. https://doi.org/10.1016/j.ijthermalsci.2021.107261

[7] Berardi, U., 2019. The impact of aging and environmental conditions on the effective thermal conductivity of several foam materials. Energy 182, 777–794. https://doi.org/10.1016/j.energy.2019.06.022

Reviewer 2 Report

The main goal is not clear from the abstract.

Keywords are too long.

The issue of container housing (or temporary housing) is interesting for today's society. However, similar container elements are already known in Europe in the second half of the last century. Structurally, these container elements are developed at a good level, for example in Europe; after 2000, thermal technical properties are gradually improved, with a link to thermal technology and energy. Therefore, the article lacks a broader comparison of how similar issues are handled in other countries or continents. A wider comparison is desirable.

Discussion, conclusion, it would be appropriate to expand on the direction in which the type of container housing will continue to develop.

Author Response

Detailed responses to reviewers’ comments of manuscript ijerph-2039407

Analysis of thermal insulation thickness for container house in Yanqing Zone of Beijing 2022 Olympic and Paralympic Winter Games

Dear Editor and reviewers,

Thank you for your thorough review and constructive comments on our manuscript! We seriously considered the comments and carefully revised the manuscript. In addition, we have made some modifications on English expression in this revision.

All the changes in the revision were marked up using the “Track Changes”. We believe most of the comments have been addressed in the revised manuscript. The point-by-point response to the reviewers’ comments are listed below. Hope this work can be accepted to publish in International Journal of Environmental Research and Public Health.

Reviewer 2

1. The main goal is not clear from the abstract.

Thank you for this question! The main goal was added in the abstract, on page 1, from line 13 to line 16, as “Taking a kind of container house (CH) used in Yanqing zone as a model, the objective of this paper is to analyze the effect of insulation thickness on heating energy consumption and corresponding carbon emission. The effect of service life of THs on economic thickness was also discussed.” . Meanwhile, the abstract of the original manuscript was partially revised, from line 16 to line 19, as “The simulation model was developed using EnergyPlus and the heating energy consumption with different insulation materials was simulated based on the meteorological parameters of the top of Xiaohaituo Mountain (2177.5 m) and the Olympic/Paralympic Village (950 m) in Yanqing zone.”.

2. Keywords are too long.

Thank you for the comment! We have adjusted the keywords and the revised keywords are “Container house; Insulation materials; Energy efficiency; Carbon emission; Economic analysis”, as shown on page 1.

3. The issue of container housing (or temporary housing) is interesting for today's society. However, similar container elements are already known in Europe in the second half of the last century. Structurally, these container elements are developed at a good level, for example in Europe; after 2000, thermal technical properties are gradually improved, with a link to thermal technology and energy. Therefore, the article lacks a broader comparison of how similar issues are handled in other countries or continents. A wider comparison is desirable.

Thanks for this comment and suggestion! The references were complemented to compare the approaches of other countries or continents on thermal technology of temporary housing. The specific content is “While the thermal performance of temporary buildings is receiving a lot of attention, the thermal technology is also improving. Honma [18] introduced that according to the special specification of the Hokkaido government, glass wool insulation of 100 mm should be applied in the wall and ceiling of emergency temporary housing. Soga et al. [19] mentioned that different solutions on thermal technology of THs had be taken in different regions of Japan, such as increasing the thickness of insulation in Hokkaido. In the Mediterranean area, Cross-Laminated Timber (CLT) technology was applied to THs [20, 21], which provides high insulation and good air tightness. It enables building structures to avoid thermal bridging and allows achieving remarkable energy savings in the heating season [22]. For CHs, Grębowski et al. [23] proposed that the envelope can be internal insulated using glass wool or insulated with sprayed polyurethane foams. And, Berbesz et al. [24] also pointed out that proper insulation is an important aspect for container house design.”, as shown on page 2, from line 67 to line 79.

4. Discussion, conclusion, it would be appropriate to expand on the direction in which the type of container housing will continue to develop.

Thank you for providing a very interesting question! The expansion in the direction in which container houses will continue to develop was added to the “Conclusion”, on page 15, from line 472 to line 478, as “CHs are now a commonly used prefabricated element in the building industry. The CHs had achieved a variety of functional applications during the Beijing 2022 Olympic Winter Games. After the game, as sustainable legacies, the CHs have been recycled and transformed as convenience stores, reading bars, hotel rooms and so on to play an important role in urban comprehensive development and urban renewal. Although the research of this paper is based on the meteorological parameters of Xiaohaituo Mountain, the results are also appropriate for similarly climate conditions.”.

Reviewer 3 Report

The manuscript covers the techno-economic study of 4 types of insulation to achieve NZEB standard in containerised temporary housing. The manuscript is well written, and is of interest to the journal. However, a few comments are in order. Tables 7 and 8 seem to be incomplete. In addition, from figure 7 it would be interesting to note in the figure from which point of the graph the NZEB standard is met.

Author Response

Detailed responses to reviewers’ comments of manuscript ijerph-2039407

Analysis of thermal insulation thickness for container house in Yanqing Zone of Beijing 2022 Olympic and Paralympic Winter Games

Dear Editor and reviewers,

Thank you for your thorough review and constructive comments on our manuscript! We seriously considered the comments and carefully revised the manuscript. In addition, we have made some modifications on English expression in this revision.

All the changes in the revision were marked up using the “Track Changes”. We believe most of the comments have been addressed in the revised manuscript. The point-by-point response to the reviewers’ comments are listed below. Hope this work can be accepted to publish in International Journal of Environmental Research and Public Health.

Reviewer 3

The manuscript covers the techno-economic study of 4 types of insulation to achieve NZEB standard in containerised temporary housing. The manuscript is well written, and is of interest to the journal. However, a few comments are in order.

  1. Tables 7 and 8 seem to be incomplete.

Sorry for the incomplete information provided in Tables 7 and 8. And, Tables 7 and 8 were revised as well as Tables 5 and 9 were modified accordingly, as on page 11, 12 and 8. And the relevant text has also been rephrased.

  1. From figure 7 it would be interesting to note in the figure from which point of the graph the NZEB standard is met.

Thank you for the comment! Figure 7 was modified and relevant information was complemented, as shown on page 12.

Round 2

Reviewer 2 Report

the article has a substantial improvement